# Impact of COVID-19 on Uro-Oncological Patients: A Comprehensive Review of the Literature

**DOI:** 10.3390/microorganisms11010176

**Published:** 2023-01-10

**Authors:** Filippo Gavi, Paolo Emilio Santoro, Carlotta Amantea, Pierluigi Russo, Filippo Marino, Ivan Borrelli, Umberto Moscato, Nazario Foschi

**Affiliations:** 1Postgraduate School of Urology, Università Cattolica del Sacro Cuore, Largo Francesco Vito 1, 00168 Rome, Italy; 2Department of Health Science and Public Health, Università Cattolica del Sacro Cuore, Largo Francesco Vito 1, 00168 Rome, Italy; 3Department of Women, Children and Public Health Sciences, Fondazione Policlinico Universitario Agostino Gemelli IRCCS, Largo Francesco Vito 1, 00168 Rome, Italy; 4Postgraduate School of Occupational Medicine, Università Cattolica del Sacro Cuore, Largo Francesco Vito 1, 00168 Rome, Italy; 5Department of Urology, Fondazione Policlinico Universitario Agostino Gemelli IRCCS, Largo Francesco Vito 1, 00168 Rome, Italy

**Keywords:** uro-oncology, COVID-19, cancer, review

## Abstract

**Background:** The aim of this paper is to discuss the impact of COVID-19 on patients with urological malignancies (prostate cancer, bladder and upper tract urothelial cancer, kidney cancer, penile and testicular cancer) and to review the available recommendations reported in the literature. **Methods:** A review was performed, through the PubMed database, regarding available recommendations reported in the literature, to identify studies examining the impact of COVID-19 on treatment and clinical outcomes (including upstaging, recurrence, and mortality) for uro-oncological patients. **Results:** The COVID-19 pandemic dramatically changed the urological guidelines and patients’ access to screening programs and follow-up visits. Great efforts were undertaken to guarantee treatments to high-risk patients although follow up was not always possible due to recurrent surges, and patients with lower risk cancers had to wait for therapies. **Conclusions:** Physically and mentally, uro-oncological patients paid a heavy price during the COVID-19 pandemic. Long term data on the “costs” of clinical decisions made during the COVID-19 pandemic are still to be revealed and analyzed.

## 1. Introduction

As of 1 December 2022, 6.63 million fatalities and 643 million cases of COVID-19 had been documented globally [1]. SARS-CoV-2 is a virus that belongs to the b-coronavirus family. It is assumed to have originated in bats due to similarities between its genomic sequence and that of the bat coronavirus, making it the third zoonotic coronavirus to infect human populations over the past 20 years [2]. SARS-CoV-2 is extremely contagious and spreads mostly through human-to-human contact, aerosol exposure, and physical contact. The primary cause of infection is thought to be the virus-carrying respiratory droplets that infected individuals cough or sneeze, which are propelled approximately one meter and can subsequently be deposited on neighboring individuals’ oral, nasal, or ocular mucous membranes. Furthermore, asymptomatic patients may transmit the disease [3]. Because of the fast spreading of SARS-CoV-2 viruses, the pandemic had a great impact on healthcare workers both physically and mentally [4,5]. Since the first half of 2020, the COVID-19 public health emergency, which has seen the planet involved in the management of COVID-19 patients, has placed a great deal of stress on health care facilities due to the increased demand for beds and consequently the intense workload for health care workers. Health care staff involved in the emergency management network were the pillars on which the response to the SARS-CoV-2 outbreak was based, and it was therefore fundamental to preserve their physical and mental health as much as possible [6].Considering the emergency scenario, as reported in the international literature, there were increased levels of burnout in healthcare workers [7] due to doubled and/or tripled shifts without adequate rest, saturated wards, and the multitude of deaths they had to witness. The Istituto Superiore di Sanità (ISS) in Italy has recommended that healthcare professionals use preventive strategies to combat the onset of this syndrome [6]. Burnout among medical professionals has been linked to lower work performance, a rise in medical errors, interpersonal conflicts, and depression. Compared to doctors in other specialties, urologists may experience higher rates of burnout (up to 63.6 percent), particularly during residency [8]. The management of uro-oncology patients has not been made any easier by the SARS-CoV-2 pandemic. The nature of SARS-CoV-2 grants it the capacity of binding to angiotensin-converting enzyme 2 (ACE2) receptors. Target cells in this setting might be human cells expressing ACE2 [9]. Organs at high risk of viral invasion include the kidney, bladder, ileum, esophagus, and heart [10]. The SARS-CoV-2 incubation period lasts between 2 and 14 days [11]. Approximately 80% of patients in Chinese reports have mild-to-moderate disease, 13.8% will have severe symptoms, and 6.1% will develop life-threatening respiratory failure, septic shock, or multiple organ failure [12]. According to the literature, SARS-CoV-2 infection is linked to a greater mortality rate in cancer patients, particularly in those who started anticancer treatment 14 days before infection [13]. Cancer surgery for COVID-19 patients may need to be postponed due to perioperative SARS-CoV-2 infection, which raises postoperative mortality [14]. As a result, these patients may have a worse prognosis and be at a greater risk of developing COVID-19. Patients who are more susceptible, such as the elderly and several categories of oncology patients, appear to have more severe SARS-CoV-2 infections [15]. On 21 December 2020, the European Medicines Agency (EMA) authorized the first vaccine against SARS-CoV-2, called Comirnaty, developed and produced by Pfizer/BioNTech, which has a 95 % effectiveness rate against COVID-19 infection. Four more vaccines developed by Moderna (7 January 2021), AstraZeneca (29 January 2021) with name changed to Vaxzevria (25 March 2021), Janssen Pharmaceuticals (11 March 2021) and Novavax (20 December 2021) have received conditional marketing authorization from the Commission because of favorable safety and effectiveness evaluations by the European Medicines Agency (EMA). Subsequently, other vaccines have been approved and are being evaluated by the EMA at various stages. Massive vaccination campaigns have been conducted worldwide, although there is currently little information available about the COVID-19 vaccination’s efficiency and safety in oncology patients [16,17]. This is due to the fact that cancer patients were omitted from the early tests using the vaccine since those with compromised immune responses would have tampered with trial efficacy rates. The COVID-19 vaccination is generally regarded as safe in oncology patients, and both they and their close family members should receive it [18,19]. One query is whether cancer patients ought to receive a particular COVID-19 vaccination. There are no studies that directly contrast various vaccination kinds in cancer patients. As a result, it is unclear at this time which vaccines are more reliable or efficient than others. Additionally, it is unknown which vaccines will be more (or less) effective against certain of the new SARS-CoV-2 strains. [20] Remdesivir is the first medication approved for the treatment of COVID-19, but clinical guidelines are still divided over the strength of the evidence supporting its usage in the management of moderate-to-severe condition. Remdesivir is an adenosine analogue with widespread antiviral activity that was first suggested as treatment for Ebola and then as treatment for COVID-19 after reports of its activity against the RNA-dependent RNA polymerase of SARS-CoV-2 [21,22]. Although new vaccines and therapies have been introduced, COVID-19 is still a burden for frailty patients. The COVID-19 pandemic has had an effect on oncologic treatment and also on medical education, research, and the health of both patients and healthcare workers [23]. The aim of this paper is to discuss the impact of COVID-19 on patients with urological malignancies (prostate cancer, bladder and upper tract urothelial cancer, kidney cancer, penile and testicular cancer) and to review the available recommendations reported in the literature.

This paper is the second in a series of literature reviews regarding urology and public health that our research group has completed [16].

## 2. Methods

A narrative review of available recommendations reported in the literature regarding the impact of COVID-19 on patients with uro-oncologic cancers was performed.

The following search terms were used to identify eligible articles through the PubMed database:

(“COVID-19” OR “SARS-CoV-2” OR COVID OR Coronavirus) AND (Urological cancer type*).

The search was limited to articles written in English, published from the beginning of the pandemic (February 2020)) up until the search was performed (December 2022), and no other filters were applied.

### 2.1. Inclusion Criteria

Articles investigating the impact of COVID-19 on treatment and clinical outcomes (including upstaging, recurrence, and mortality) of patients with:Bladder cancer (BC);Prostate cancer (PCa);Renal cell carcinoma (RCC);Upper tract urothelial carcinoma (UTUC);Penile cancer and Testicular cancer.

### 2.2. Exclusion Criteria

Manuscripts that did not investigate the impact of COVID-19 on the treatment and clinical outcomes of patients with uro-oncologic cancers were excluded.

### 2.3. Data Extraction and Synthesis

Two researchers (F.G. and C.A.) performed data extraction from the included articles and reported the results in an Excel worksheet. One author (F.G.) extracted data from the included studies, and the second author (C.A.) checked the extracted data. Disagreements were resolved by a consensus-based discussion between the two review authors (F.G. and C.A.); if agreement could not be reached, a third author (N.F.) was scheduled to inter-vene.

## 3. Results

The research findings are reported by uro-oncologic tumor type as follows.

### 3.1. Prostate Cancer

With the explosive increase of patients with COVID-19, unprecedented limits were placed to stop the spread of SARS-CoV-2 and to reorganize the healthcare system and prevention campaigns as a whole [24]. The complete range of cancer screening and diagnosis gaps have probably been brought on by these changes. Prostate cancer (PCa) screening and diagnosis programs were temporarily delayed despite the guidelines [25]. PSA testing has been an affordable, minimally intrusive, and reasonably accurate method of PCa screening that would increase the detection of any kind of PCa. Sadly, a decline in PSA screening would result in a much lower rate of PCa detection and a non-negligible increase in deaths specifically from PCa [26,27,28].

The first year of the COVID-19 pandemic has changed our urologic practice dramatically for both patients with benign and malignant diseases [29,30]. The majority of urologic societies advised patients with prostate PCa to postpone any surgical therapy [31,32]. Oncologic results for PCa patients may be negatively impacted by a delay between diagnosis and surgical intervention (i.e., surgery delay). Diamand et al. [33] collected and retrospectively analyzed the data of 926 patients undergoing radical prostatectomy for localized intermediate- and high-risk PCa. In their study, there was no evidence that a radical prostatectomy’s oncologic results were negatively impacted by the delay caused by the pandemic. Researchers found no connection between surgical hold-up and unfavorable oncologic outcomes, such as biochemical recurrence, pathological locally advanced illness, upgrading, or need for adjuvant therapy. Such results are in line with a previous article published by Gupta et al. [34] a year before the COVID-19 pandemic regarding the impact of length of time from diagnosis to surgery in patients with localized prostate cancer. In light of the COVID-19 pandemic, the literature indicates that “postponed strategy” for PCa patients who are waiting for surgery is safe [33]. A significant decrease in radical prostatectomy was registered in a Turkish multicenter study [35] and the pathological Gleason score was higher in patients who underwent surgery during the COVID-19 pandemic in comparison to previous years. No data regarding the impact of such results were further published. A novel microsimulation model was developed by Ward et al. [36] to predict the effects of COVID-19 on excess mortality and cancer detection by month throughout the pandemic, as well as anticipated cancer diagnoses, outcomes of stage at diagnosis, and survival through 2030 [36]. The data suggested that in addition to a significant anticipated increase in cancer diagnoses, delays in diagnosis will result in a worse disease stage at presentation, which will negatively impact survival rates [36].

Pharmacological treatments for prostate cancer include androgen deprivation therapy (ADT). In recent years, articles on how ADT can help to prevent or lessen the severity of COVID-19 were published [37,38,39]. Bahmad et al. [40] proposed a possible mechanism that could explain why ADT has a protective role against COVID-19 (Figure 1). However, the results in the literature are contradictory. The differences in patient selection and the fact that they did not examine several crucial clinical and demographic factors that should be considered when evaluating the interaction between ADT and SARS-CoV-2 infection outcomes limit the comparability and interpretation of the available studies. Larger and more accurate investigations are required, taking into account factors that alter infection outcomes, as these have a substantial impact on the validity of the results [41].

Bernstein et al. [42] conducted a multicenter study to assess the association between race and PCa treatment in patients with nonmetastatic PCa during the COVID-19 pandemic in the USA. In this large multi-institutional regional collaborative cohort study, Black patients had lower likelihood of having PCa surgery during the first wave of the COVID-19 pandemic than White patients. Although localized PCa does not need to be treated right away, this study’s results show that there were systemic disparities in the healthcare system in USA during the COVID-19 pandemic.

### 3.2. Bladder and Upper Tract Urothelial Cancer

Hematuria is a worrisome symptom and can be a sign of bladder cancer. During COVID-19, its management is even more challenging. Although there is no widely acknowledged standard procedure, visible hematuria always needs to be investigated because its presence indicates a risk of roughly 20.4% malignancy, compared to 2.7% in the case of microscopic hematuria [43,44,45]. Additional risk factors should be considered in patients with hematuria and can be useful to stratify them, prioritizing those with a higher risk of cancer. The IDENTIFY collaborative study aims to develop a prediction model for urinary tract cancer in patients with hematuria, and it has already showed promising results [44]. Lower urinary tract symptoms (LUTS) could be the first sign of bladder cancer [46]. LUTS have a great impact on the quality of life of patients, and they need to be assessed and managed by a multidisciplinary team [47,48]. During the COVID-19 pandemic, non-urgent visits were suspended and some causes of LUTS were not deeply investigated. The preferred diagnostic procedure for bladder cancer is still a cystoscopy, and when unequivocal lesions are seen on an ultrasound or computed tomography of the urogram, it is advised to perform a transurethral bladder resection (TURB) [49,50].

Cystoscopy is still the gold standard for the diagnosis of bladder cancer, although in the worst months of the COVID-19 pandemic, cystoscopy appointments were cancelled for most patients. An important factor delaying diagnosis or therapy may be a patient’s psychological state and fear of contracting COVID-19. Delaying routine treatment could inevitably result in progression or recurrence for patients with bladder cancer or suspected bladder cancer who require long-term surgical management and have locally advanced tumors or rapid tumor growth. Urologists should not stop screening since bladder cancer has a far higher fatality rate than COVID-19, which can be as high as 52% [45].

Bladder cancer can be classified in four categories:


*1. Low-grade non–muscle-invasive bladder cancer (NMIBC)*


NMIBC, or low-grade non-muscle-invasive bladder cancer, is a condition that is often not aggressive. Active surveillance (AS) is a key management strategy for recurrent low- and intermediate-risk NMIBCs, which have long-term BC-specific death rates of approximately 1–2%. If the patient is recurrence-free, guidelines recommend discharge after 12 or 60 months [51].


*2. High-grade NMIBC*


In high-grade NMIBC, 15–40% of patients proceed to muscle invasion/metastases, and 10–20% of people may pass away from BC. Radical cystectomy (RC) and immunotherapy with bacillus Calmette-Guérin (BCG) are the main therapies [43]. Up to 8% of initial pTa tumors and 32% of pT1 tumors exhibit muscle invasion with early resection. Patients with re-resections that do not include tumors have decreased progression risks (approximately 10% every five years) [52].


*3. Muscle-invasive bladder cancer*


Several studies showed that delays between neoadjuvant chemotherapy and radical chemotherapy were connected to poor survival results, while another research showed that delays were connected to upstaging [52].


*4. Advanced or metastatic BC*


For the majority of patients with advanced or metastatic BC, cytotoxic chemotherapy with or without immunotherapy continued to be the preferred course of action.

The European Association of Urology (EAU) updated their guidelines according to the current COVID-19 pandemic [31]. Delaying a cystoscopy by six months for individuals with low priority bladder cancer who have low or intermediate risk non-muscle invasive bladder cancer (NMIBC) and no hematuria [31]. Patients with intermediate priority who have a history of high-risk NMIBC but no hematuria may be monitored before the end of three months. High priority patients who have NMIBC and sporadic hematuria must have a follow-up cystoscopy within six weeks or less [31]. Cystoscopy or TURB should be considered in cases of exigencies (visible hematuria with clots, urine retention) within fewer than 24 h. When TURB is required, special care must be made to ensure a muscle sample, preventing the need for another surgical procedure and a hospital stay. However, when TURB is required, owing to the elevated pace of disease progression, it should not be delayed [31].

CT urography is still essential to investigate possible upper urinary tract urothelial cancer (UTUC). Focused research on UTUC has shown that treatment rescheduling resulted in poorer prognostic outcomes overall, including worse pathologic staging, the presence of carcinoma in situ (CIS), tumoral infiltration, and other factors [53,54]. The UTUC risk profile should be used to determine the timing of the intervention. For instance, when surgery is put off for longer than a month, ureteral tumors have a poorer prognosis than their renal pelvic counterparts [54]. However, when surgery was postponed for more than three months, individuals with pT2 or higher UTUC did not exhibit inferior survival results [53,54]. The findings for metastasis-free and recurrence-free survival were the same [55]. The EAU guidelines were the only source that addressed follow-up and advised patients with a history of high-risk NMIBC to delay upper tract imaging by six months (low priority) [31].

The EAU advised that regimens of intravesical bacillus Calmette-Guérin treatment lasting more than a year might be safely stopped for patients with high-risk NMIBC during the COVID-19 pandemic. Moon et al. [56] reported that during the COVID-19 pandemic, there were no differences between patients receiving BCG maintenance therapies and patients receiving BCG induction therapy in terms of recurrence rate.

Despite the guidelines’ updates, during the COVID-19 pandemic, routine oncological follow up controls and screening procedures and elective surgeries were deprioritized [57]. For individuals with bladder cancer as their primary diagnosis, worse histopathological results were seen, leading to a rise in the incidence of advanced and more aggressive tumor stages [57]. In the treatment of patients with primary bladder cancer, earlier surgical treatments with precise histological staging should be taken into consideration, regardless of the risk of a potential COVID-19 exposure [57]. This is an issue, especially for patients receiving neoadjuvant chemotherapy because a delay of more than three months may have a detrimental effect on the prognosis [58]. Some patients with muscle-invasive bladder cancer might even forego surgery in favor of less effective therapies, such as radiation and chemotherapy [58]. Delay in surgical treatment could results in worse pathological and oncological results when it comes to MIBC. For instance, waiting longer than 10 weeks before having a radical cystectomy was linked to poor pathological (upstaging, positive nodal status, and positive surgical margins) and survival results [59]. In this context, both surgical margins status and locations [60] and positive nodal status are essential factors to take into account to delineate the best management decision [61]. The expected increase in bladder cancer in the next months is one aspect of COVID-19′s effect on worldwide public health. Delays in cancer diagnosis and treatment during the pandemic increase the likelihood that thousands of cases would go unrecognized and untreated, leading to an increase in untreated bladder cancer incidence in the months to come [62].

### 3.3. Kidney Cancer

The expression of the ACE2 and transmembrane protease serine 2 (TMPRSS2) receptors is upregulated in the urinary system, particularly in the kidneys, designating these organs as crucial targets for SARS-CoV-2 infection. Although mesangial cells and glomerular endothelial cells do not express ACE2 or TMPRSS2, they are expressed by proximal tubular cells and, to a lesser extent, podocytes [10,63]. AKI was a presenting symptom in 25% of COVID-19 patients. The lung-kidney axis, which is brought on by an excess of the hormone IL-6, is most likely the primary organ cross talk implicated in the pathogenesis of AKI induced by COVID-19 infection [64]. Systemic consequences often follow clinical issues in critically ill ICU patients [65].

Patients with localized renal cell carcinoma (RCC) are often treated with resection, radiographic monitoring, and adjuvant immunotherapy, depending on the tumor stage. During imaging procedures, patients under radiographic monitoring may be exposed to COVID-19 [66,67]. Patients with advanced tumors are treated systemically with targeted drugs, immunotherapy, or a combination of the two [68].

According to some studies, immune checkpoint inhibitor therapy exposure may be a separate risk factor for the emergence of a more severe clinical course of COVID-19 infection. This may be because immune checkpoint inhibitors exposure may cause immune-related pulmonary toxicities and increased T-cell cytokine production [69]. Contrarily, several recent investigations in a range of cancer types have not revealed an appreciable increase in the probability of contracting or dying from a COVID-19 infection while taking antiprogrammed cell death protein-1 (PD-1) immune checkpoint inhibitors [70].

Some retrospective investigations have found no correlation between acquiring immune checkpoint inhibitors and having a COVID-19 infection, and no increase in the severity or mortality of the infection [71]. No data is currently available that shows the use of immune checkpoint inhibitors enhances the risk or severity of a COVID-19 infection, hence conventional guideline-based approaches to treatment are still advised when using systemic therapy techniques. One of the immune checkpoint inhibitors’ adverse events is pneumonitis [72]. In order to correctly and quickly treat a potentially fatal adverse event, it is crucial to constantly monitor patients on these regimens for such occurrences and to rule out a COVID-19 infection as soon as symptoms appear. Additionally, in some clinical situations, it is not unreasonable to consider permanently stopping immune checkpoint inhibitors therapy for a patient with RCC who is receiving immune checkpoint inhibitors therapy should they experience any life-threatening adverse events or are thought to be at high risk of experiencing them after 2 years of treatment [73].

In the face of COVID-19, doctors may have suggested de-escalating therapy and monitoring according to the EAU guidelines [67]. During the COVID-19 surge, resources were reallocated. Patients with RCC were triaged in low, intermediate, and high priority. In a study conducted in a London hospital, data of 426 patients with RCC were retrospectively analyzed. In London, COVID-19 had a significantly negative impact on acute and elective care during the initial peak in spring 2020. By postponing low-priority patients during the surge, it was intended to free up surgical capacity for high- and intermediate-priority cases. In this particular retrospective study, no immediate clinical impact was seen thanks to prioritizing and COVID-protected pathways, which preserved the ability for time-sensitive oncological therapies [74].

Apparently a delay of less than 3 months in surgical treatment of patients with a diagnosis of T1b or T2b renal cell carcinoma did not significantly increase the risk of tumor progression and change of surgical approach [75]. Wai-Shun Chan et al. [76] published a systematic review and meta-analysis, including 11 studies of quantitative analysis, affirming that there was insufficient evidence to support the notion that delayed surgery is safe in localized RCC. For metastatic RCC, upfront targeted therapy followed by deferred cytoreductive nephrectomy should be considered [76]. Current stratification systems for selecting patients for cytoreductive nephrectomy in metastatic disease are controversial. It is not always clear which patients should undergo upfront surgery instead of systemic oncological treatment. In that scenario the REMARCC risk score has been developed to help surgeons in selecting optimal candidates for cytoreductive nephrectomy among patients with metastatic renal cell carcinoma, although further prospective external validations are still required [77].

Other studies reported a significant decrease in diagnostic procedures and treatment, the consequences of which are still to be investigated [78,79]. Given the recurrent surges that Europe and the UK have seen, certain low-priority patients with small kidney tumors may have been in danger of being delayed for a longer period of time than is acceptable [80]. It is possible that the true mid- and long-term “costs” and effects of the COVID-19 pandemic on the management of RCC are still unknown.

Moreover, Khene et al. [81] demonstrates a concerning decline in public awareness of urological malignancies during the COVID-19 pandemic, which might start a vicious cycle that has negative impacts on individuals, healthcare systems, and society as a whole. This is critical because a decline in interest might ultimately lead to a decline in awareness, which could have clinically significant effects on patient compliance with early detection and/or screening routes for urological malignancies. In order to achieve early detection and value-based care for illnesses, including prostate cancer, bladder cancer, and kidney cancer, patient empowerment is essential [82,83].

### 3.4. Penile Cancer

Penile cancer is a rare condition, representing 1% of all cancers in men. The highest incidence is between 60 and 80 years of age. Due to the low incidence of penile cancers, few resources were invested during the COVID-19 pandemic [84]. In order to reduce hospital visits during the COVID-19 pandemic, some experts concurred that individuals with penile lesions that are clinically visible malignancies should not obtain a biopsy to confirm the diagnosis before beginning a final course of therapy in order to decrease the frequency of hospital visits [85]. More crucially, individuals with penile lesions that have a low index of suspicion should not have biopsy but rather should be kept under monitoring. In order to assess the preoperative disease extent; all patients with a new diagnosis of penile cancer should merely go through a physical examination rather than stimulated penile magnetic resonance imaging (MRI, which requires an injection of alprostadil for stimulation) [85].

On the other hand, delaying penile cancer therapy might cause the underlying tumor’s disease to advance to the point where organ-preserving surgery may no longer be an option for treating the disease since penile cancer patients typically wait too long to be attended [86]. Consensus recommendations advise that patients with advanced or metastatic cancer be given the option of neo-adjuvant or adjuvant chemotherapy with or without radiation therapy [85]. All guidelines recommended paying close attention to inguinal lymph nodes involvement since they have an impact on prognosis. Treatment must not be postponed in such cases [31,87].

### 3.5. Testicular Cancer

Testicular cancer is an uncommon neoplasm that accounts for 1–2% of tumors in men. In most cases, it is curable, but it has the highest incidence in young adults (14–34 years) [88]. One testicle will often expand without any discomfort as a symptom of testicular cancer, which is frequently accidentally discovered. Metastasis have been described in 10% or less of patients at diagnosis. Due to the fast growth of testicular tumors—which double in size every 20 to 30 days—early identification and treatment of testicular cancer are essential for the best chance of survival [89].

Studies conducted before the COVID-19 pandemic have shown a significant increase of tumor progression in 30 days between early orchiectomy group and delayed orchiectomy group [90]. Radical inguinal orchiectomy combined with or without retroperitoneal lymph node dissection is a common early therapeutic option for patients with testicular cancer. With the development of effective chemotherapy regimens, there is no differences in survival rates between early and delayed orchiectomy [90].

Seminomas are usually radiosensitive and chemo-sensitive, while non-seminomas are less sensitive to radiation. In case of metastatic disease, chemotherapy and surgery are usually required [86,88,89].

There is not as much data on the impact of postponing surgical therapy due to COVID-19 for testicular cancer. According to a recent analysis, testicular cancer patients would benefit from minimizing delays and their management should be prioritized [91].

## 4. Discussion

The healthcare system is under tremendous strain because of the COVID-19 epidemic. As of 1 December 2022, 6.63 million fatalities and 643 million cases of COVID-19 had been documented globally [1]. Uro-oncological patients are a particularly vulnerable population since they frequently spend a lengthy time in hospitals, which increases their chance of contracting COVID-19 [20]. We reviewed the available literature on how the COVID-19 pandemic changed the approach to urological cancers and how that change could have affected uro-oncological patients. Few data reported an increase in mortality, missed diagnosis or delayed diagnosis due to the suspension of screening test, programmed follow up or elective treatment during the COVID-19 pandemic. Several authors have suggested that the real “costs” of such decisions are still to be seen and analyzed. The use of different levels of prioritization seemed to have helped during the worst months of the COVID-19 pandemic to allocate the scarce resources on the most urgent situations, although few data regarding the long term have been reported. Use and evolution of telemedicine (remote monitoring, phone calls or video conference) have helped physicians to counsel patients in a way that was not possible before [92]. It is worth remembering though that telemedicine is not available to all. Elderly and frailty patients could need an easier platform to be able to use such a useful tool in time of need. Vaccinations and implementation of COVID-19 pathways were essential to guarantee elective surgeries and follow up to uro-oncological patients [93]. Among the groups of people most susceptible to developing severe types of COVID-19 are oncological patients. According to reports published by ESMO guidelines and several studies in the relevant literature [94,95], mortality rates among cancer patients with SARS-CoV-2 infection can range from 5 to 61 percent, significantly higher than those observed in the general population. As our review showed, the pandemic has affected the frequency of screening and diagnosis programs, sometimes with delays in identifying cancers that are themselves a cause of increased mortality. Changes have also occurred in treatment programs, with changes in the timing of administration of some drugs, delays in the initiation of therapies or the performance of surgeries. In addition, for this reason, cancer patients were considered a “priority category” to receive the COVID-19 vaccination, particularly during the early stages of the vaccination campaign (which started across Europe on 27 December 2020 with the so-called “Vaccine day”) when the available vaccine doses were insufficient to cover the entire population. The COVID-19 immunization is strongly recommended for oncological patients, according ESMO guidelines [96] due to their extremely elevated risks of infection. The latest information in the literature suggests that COVID-19 vaccination safety in cancer patients is comparable to that seen in the general population. Additionally, as with the general population, even for cancer patients, the benefits of vaccination far outweigh the risks. All SARS-CoV-2 vaccines have been proved to be effective, but mRNA vaccines are thought to be highly effective and secure and are ideally suited for patients with a damaged immune system or who are receiving chemotherapy.

## 5. Conclusions

The diagnosis of cancer has a great psychological burden on patients. During the COVID-19 pandemic, it was likely magnified due to uncertainty and treatment delays. Physicians had to reallocate resources and prioritized treatment as the guidelines suggested, considering age, comorbidities, symptoms, and life expectancy. Physically and mentally, uro-oncological patients paid a heavy price during the COVID-19 pandemic. Finally, despite scientific societies efforts to rewrite guidelines for the pandemic period, all oncology patients suffered from delayed controls, treatments and diagnosis. Long term data on the “costs” of such decisions are still to be revealed and analyzed.

## Figures and Tables

**Figure 1 microorganisms-11-00176-f001:**
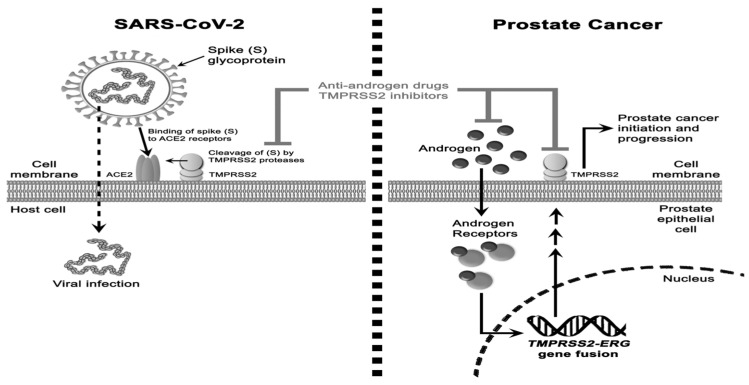
The interaction between COVID-19 and prostate cancer proposed by Bahmad et al. [40].

## Data Availability

Data are available upon reasonable request.

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
