# Peer review of "Impact of COVID-19 on Uro-Oncological Patients: A Comprehensive Review of the Literature"

_microorganisms, 2023, doi:10.3390/microorganisms11010176_

Round 1
Reviewer 1 Report
The paper is a comprehensive review of health literacy and vaccination intent and status. The topic is highly significant and of great interest to readers given the recent experiences with the COVID-19 pandemic and the delay involving the management of numerous diseases. The abstract is well written, and the cited references are appropriate and up to date. Below are some suggestions of clarifications and additions to the authors.
1 - The Evidence acquisition section should be renamed as methods section
2 - The Evidence synthesis section should be renamed as Results section
3 - The methods sections should be improved with an explicit description of inclusion and exclusion criteria and of data extraction. If appropriate for the kind of papers retrieved, a quality assessment should be performed.
4 - The results section should report the total number of papers retrieved by the research string and the total number of papers included in the review. A PRISMA flow chart should be included, as well as a table with the characteristics of the papers included.
5 - Throughout all the results sections, it’s difficult to distinguish between results from retrieved paper and authors consideration. All authors considerations should be moved to the discussion section.
6 - Throughout all the results sections, sentences should be conveniently referenced at their conclusion, and not at the end of the paragraph only.
7 - The discussion section should be more focused on the findings of the review and on the consequences in patients’ management.
Author Response
We thank the reviewer for very interesting comments. We have responded point by point as follow:
1 - The Evidence acquisition section should be renamed as methods section.
Answer: The name of this section was changed accordingly.
2 - The Evidence synthesis section should be renamed as Results section.
Answer: The name of this section was changed accordingly.
3 - The methods sections should be improved with an explicit description of inclusion and exclusion criteria and of data extraction. If appropriate for the kind of papers retrieved, a quality assessment should be performed.
Answer: We thank the reviewer for the comment. We've made it clearer in the "Methods" section that this is a "narrative review of the literature," (following the editor's request for the special issue) and as suggested, we have better clarified the inclusion, exclusion, and data extraction criteria. We haven’t done a systematic review because we were invited for the submission of a narrative review and not a systematic review.
4 - The results section should report the total number of papers retrieved by the research string and the total number of papers included in the review. A PRISMA flow chart should be included, as well as a table with the characteristics of the papers included.
Answer: Thanks for the interesting point. As clarified in the methods, we carried out a narrative review of the literature, therefore, we did not follow a standardized methodology such as the PRISMA guidelines (Page, M.J.; McKenzie, J.E.; Bossuyt, P.M.; Boutron, I.; Hoffmann, T.C.; Mulrow, C.D.; Shamseer, L.; Tetzlaff, J.M.; Akl, E.A.; Brennan, S.E.; et al. The PRISMA 2020 Statement: An Updated Guideline for Reporting Systematic Reviews. BMJ 2021, 372, n71.)
5 - Throughout all the results sections, it’s difficult to distinguish between results from retrieved paper and authors' consideration. All authors' considerations should be moved to the discussion section.
6 - Throughout all the results sections, sentences should be conveniently referenced at their conclusion, and not at the end of the paragraph only.
7 - The discussion section should be more focused on the findings of the review and on the consequences in patients’ management.
Answer 5,6,7: We thank the reviewer for the comment, and we have implemented the citation in order to clarify which parts were the authors’ opinions or the papers’ results and discussion according to his suggestions.
Reviewer 2 Report
In this interesting paper the authors sought out to clarify the impact of COVID pandemic on urological malignancies.
I have no major concerns: the paper reads well. Evidence acquisition was clearly described in the appropriate section.
I would enrich the chapters about bladder cancer renal cell carcinoma.
In general, huge information about the haematuria management were provided by the Identify Study and its predictive model. The authors should consider to add these references in this specific context (doi: 10.1111/bju.15483; doi: 10.1016/j.euf.2022.06.001.).
Considering MIBC, delay in surgical treatment could be translated in worse pathological - and then - oncological outcomes. For example a delayed time to radical cystectomy greater than 10 weeks after NAC administration was associated with adverse pathological (upstaging, positive nodal status, positive surgical margins) and survival outcomes (doi: 10.1016/j.euo.2018.09.004) requiring further treatments. In this context, the importance of both surgical margins status and location (doi: 10.1007/s00345-021-03776-5) and positive nodal status (doi: 10.1016/j.euo.2022.04.001) were highlighted in a large cohorts of RC candidates.
Considering renal cell carcinoma, it must be pointed out the rising number of metastatic de nove RCC suitable of cytoreductive nephectomy. In this context the risk score coming from the REMARCC consortium to select best candiate for upfront surgery instead of oncological treatment is worth mentioning (doi: 10.1016/j.euo.2020.12.010).
Author Response
Answer: Dear reviewer, thank you for your precious comments and references. As you can see we have cited all the references you suggested and completed both the bladder cancer and the renal cell carcinoma paragraphs. Thank you again!
Round 2
Reviewer 1 Report
All comments have been adequately addressed, i'm pleased to recommend your paper for publication.